

# Assessment of the ecotoxicity of urban estuarine sediment using benthic and pelagic copepod bioassays

Maria P. Charry[1,2], Vaughan Keesing[3], Mark Costello[4] and Louis A. Tremblay[1,2]

[1] School of Biological Sciences, University of Auckland, Auckland, New Zealand
[2] Cawthron Institute, Nelson, New Zealand
[3] Boffa Miskell, Wellington, New Zealand
[4] Institute of Marine Science, University of Auckland, Auckland, New Zealand

## ABSTRACT

Urban estuarine sediments are sinks to a range of contaminants of anthropogenic origin, and a key challenge is to characterize the risk of these compounds to receiving environments. In this study, the toxicity of urban estuarine sediments was tested using acute and chronic bioassays in the benthic harpacticoid *Quinquelaophonte* sp., and in the planktonic calanoid *Gladioferens pectinatus*, two New Zealand copepod species. The sediment samples from the estuary tributary sites significantly impacted reproduction in *Quinquelaophonte* sp. However, results from one of the estuary sites were not significantly different to those from the tributaries sites, suggesting that chemicals other than trace metals, polycyclic aromatic hydrocarbons and ammonia may be the causative stressors. Sediment elutriate samples had significant effects on reproductive endpoints in *G. pectinatus*, and on the induction of DNA damage in cells, as shown by the comet assay. The results indicate that sediment contamination at the Ahuriri Estuary has the potential to impact biological processes of benthic and pelagic organisms. The approach used provides a standardized methodology to assess the toxicity of estuarine sediments.

## INTRODUCTION

Population growth in coastal areas has resulted in the degradation of estuarine habitats and loss of biodiversity (*Lotze et al., 2006*; *Micheli et al., 2013*). Over 40% of the worldwide population lives within 100 km of the coast, and in New Zealand, 75% of human settlements are within 10 km of the coast (*Sale et al., 2014*; *Statistics New Zealand, 2006*). The growing demand for resources and services has increased the pressure for agricultural and industrial activities, and urban development (*Neumann et al., 2015*; *van Vliet et al., 2015*). Estuaries have become highly vulnerable to this pressure, due to the continuous input of municipal, agricultural and industrial runoff, storm water discharges and accidental wastewater overflows (*de los Ríos et al., 2016*; *Risch et al., 2018*; *Rodrigues et al.,*

Corresponding author
Maria P. Charry,
maria.charry@cawthron.org.nz

2017; *Willis et al., 2017*). These stressors, added to the increase in sedimentation, have made estuaries a sink for pollutants with affinity for small particles (*Reichelt-Brushett, Clark & Birch, 2017*; A. Swales et al., 2012, unpublished data: report HAM2012-048, available upon request from NIWA; *Vermeiren, Muñoz & Ikejima, 2016*).

The potential risk of traditional pollutants and emerging contaminants has become a concern for environmental regulators across New Zealand. The Ahuriri Estuary is recognized as a significant conservation area under the Regional Coastal Environmental Plan for Hawke's Bay (*HBRC, 2014*), and is surrounded by 175 hectares of wetlands of ecological importance. The Purimu Stream and the County Drain were created as buffering zones to reduce stormwater pollution at the Ahuriri Estuary. *Smith (2014)* reported the presence of contaminants above threshold limits in the Ahuriri Estuary. Recent studies using in vitro and zebrafish (*Danio rerio*) embryo tests showed that sediment extracts had genotoxicity, teratogenicity and acute toxicity (*Boehler et al., 2017*; *Heinrich et al., 2017*).

Studies on chemical extracts can identify chemicals and provide insights into the mechanisms of toxicity but their ability to predict bioavailability or impact on whole organisms is limited (*Burton, 1991*; *Gyuricza, Fodor & Szigeti, 2010*). Whole sediment assessments applying in vivo and in vitro approaches can complement extract studies by providing information closer to real field situations. The test organisms are exposed to complex chemical mixtures that can lead to synergistic, additive, or antagonistic toxicities. This is important as not considering toxicity of mixtures can result in under or over-estimation of the environmental risk (*Crain, Kroeker & Halpern, 2008*; *Deruytter et al., 2017*; *Hasenbein et al., 2015*; *Przeslawski, Byrne & Mellin, 2015*).

Other endpoints can be measured to further characterize the mechanisms of toxicity like oxidative DNA damage that can lead to genotoxicity (*Esperanza et al., 2015*; *Frenzilli, Nigro & Lyons, 2009*; *Pellegri, Gorbi & Buschini, 2014*). This can be measured by the comet assay, a single cell electrophoresis technique in which the DNA supercoil is relaxed and exposed to electrophoresis, allowing DNA with single and double breaks to migrate towards a charged anode. The amount of DNA visible in the comet tail under fluorescence microscopy provides an estimate of the extent of damage in a cell (*Azqueta & Collins, 2013*). DNA damage has been reported to affect enzyme functioning and impair the immune system and metabolism of invertebrates, resulting in growth, development, and fitness alterations (*Bajpayee, Kumar & Dhawan, 2017*; *Buschini et al., 2003*). Genetic alterations can also lead to mutations and cell proliferations (*Mussali-Galante et al., 2014*; *Simonyan et al., 2016*), and inherited teratogenic defects, leading to decreased fitness in populations (*Bachère et al., 2017*; *De Flora, Bagnasco & Zanacchi, 1991*; *Osman, 2014*).

The aim of this study was to characterize the toxicity of urban sediment using benthic and pelagic copepod species. The approach combined chemical analysis, bioassays on sediment samples and their elutriates, and the comet assay to estimate genotoxicity potential.

# MATERIALS AND METHODS

## Sampling sites and sediment preparation

The Ahuriri Estuary is located north of the city of Napier (population 60,000), on New Zealand's North Island (39°30′S, 176°52′E) as described previously (*Heinrich et al., 2017*).

Four sites were selected: two were located 50 m upstream from the Estuary at the Old Tutaekuri Riverbed, and at the Humber Street that is exposed to the Tyne Street stormwater drain network. The other two samples were collected downstream from the tributaries sites, at the Ahuriri Estuary (Fig. 1). The Waitangi estuary was used as the reference site control (based on previous chemical analysis results), and samples were collected from the mouth of the Ngaruroro river. All samples were collected in August and September 2017, and stored at 4 °C for two days prior to use two replicates of sediment cores were collected for each site by scraping the top 0–5 cm surface sediment. Sample replicates from each site were pooled, and split into subsamples for metal and organic analyses and bioassays. The pH, salinity and Redox of the samples were measured using a multiparameter probe (Hach HQD meter field No.58258.00).

**Test species**

*Quinquelaophonte* sp. (M.P. Charry et al. 2018, unpublished data) is a benthic harpacticoid copepod, native to New Zealand coastal zones. Its geographic range expands from silty sediments in the Houhora Harbour (North Island), to silty muddy sediments in Portobello Bay in the Otago Harbour (South Island). *Gladioferens pectinatus* (*Bayly, 1963*) is a pelagic species of calanoid copepod commonly found in New Zealand, Australia and Tasmania (*McKinnon & Arnott, 1985*). *G. pectinatus* is highly tolerant to salinity fluctuations, allowing it to distribute from freshwater lakes, estuaries and coastal areas, to open waters (*Bayly, 1965*; *Hall & Burns, 2002*). Both species were collected, isolated and grown in monocultures in an artificial salt water reticulation system, with controlled temperature (20 °C), salinity (30 ppt), light:dark photoperiod (12:12 h), light intensity (10–15 μmol) and dissolved oxygen (>7 mg · L$^{-1}$), following the culturing methods described in *Stringer et al. (2012)*, *Chandler & Green (1996)* and *ISO (2015)*. Dietary requirements were modified for *G. pectinatus*, following *Payne & Rippingale's (2000)* study. Cultures of *Quinquelaophonte* sp. and *G. pectinatus* were fed twice per week with a mixed algae diet of $2 \times 10^6$ and $5 \times 10^5$ cells · mL$^{-1}$ of *Isochrysis galbana*, *Chaetoceros muelleri* and *Dunaliella tertiolecta* respectively. The three species of microalgae were grown in F2 media at Cawthron Institute.

**Sediment toxicity test**

Whole sediment bioassays were conducted following methods described by *Chandler & Green (1996)*, with a modification for *Quinquelaophonte* sp. (*Stringer et al., 2014*). The assay was conducted over 14 days on a semi-static system, with a 14/10 light/dark photoperiod, temperature 20 ± 2 °C, water salinity of 30 ± 1 ppt, pH 8 ± 0.2 and DO >7 mg · L$^{-1}$. For each treatment there were four replicates. Three replicates were used for biological analysis, and one replicate for physicochemical analyses. Test chambers consisted of 50 mL borosilicate Erlenmeyers, with one diameter apertures in the neck of the flask, covered with a 55 μl nylon mesh. These openings allowed for continuous water circulation. Each flask contained 10 g of sediment, 15 adult males and 15 non-gravid females, previously isolated via glass pipette. Copepods were fed every third day with $2 \times 10^6$ cells · mL$^{-1}$ of an algae mix of *I. galbana*, *Chaetoceros muelleri* and *Dunaliella tertiolecta*. Water parameters
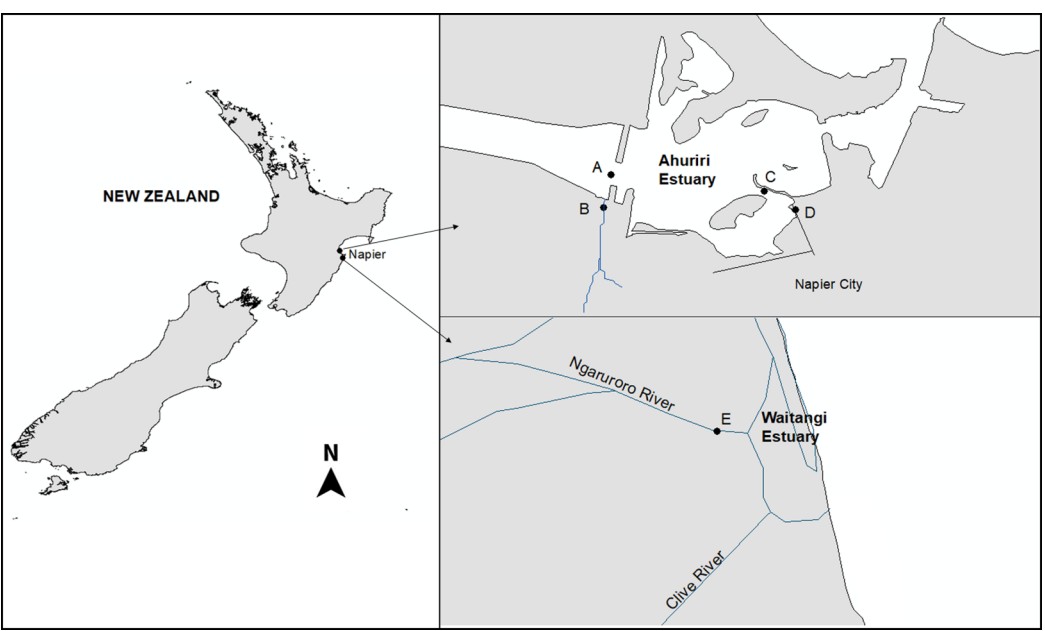

**Figure 1 The Ahuriri Estuary and Waitangi Estuary sampling sites.** (A) Old Tutaekuri Estuary site. (B) Old Tutaekuri Riverbed site. (C) Humber Estuary site. (D) Humber Drain site. (E) Waitangi Estuary site. The Old Tutaekuri Riverbed receives most of the inflow from a rural and urban catchment, and the Humber Drain receives stormwater inflow from the Pandora industrial area and residential suburbs.

were measured at the beginning and end of tests, and endpoints were survival and potential and realized offspring.

## Elutriate preparation and toxicity test

The six-day elutriate bioassays used *G. pectinatus* neonates (<12 h old). Elutriates were prepared following *ASTM E (2014)* guidelines. Briefly, sediments were refrigerated over 24 h, while wet and dry samples were weighed to determine wet weight/volume. Sediment was mixed with artificial salt water (32 g of Red Sea® Sea salt per 1 L of 0.22 μm filtered deionized water, pH adjusted to 8 ± 0.2) to reach a 1:4 (v/v) sediment: water ratio. Samples were placed in an orbital shaker at 100 rpm for 1 h, and then centrifuged at 3,000 rpm for 10 min at 4 °C. When necessary, elutriates were filtered with a 150 μm sieve, and pH was adjusted to 8 ± 0.2. All test solutions were prepared in flasks previously acid washed with 10% nitric acid for 24 h, and thoroughly rinsed with deionized water.

The ISO/CD 16778 standard (*ISO, 2015*) was followed to assess chronic effects on copepods survival and larval development. Modifications were employed to the protocol due to the hatching difference between *G. pectinatus* and the standard organism, *Acartia tonsa*. Gravid females were isolated and monitored every 4 h to ensure nauplii used were in the first larval stage (<12 h from hatching). Neonates were isolated and exposed to each treatment in triplicates. Copepod chambers were maintained at 20 ± 2 °C and under a 14:10 light dark photoperiod. Fresh elutriates were prepared and exchanged in test chambers every third day, and nauplii fed a mixture of algae ($5 \times 10^4$ cells · mL$^{-1}$) on the same day. Water samples were collected at the beginning of the test for chemical analysis.

At the end of the test, naupliar development and survival were assessed by calculating larval development ratio (realized copepodite/number of copepodites + number of nauplii) and mortality.

## Comet assay

Genotoxicity was evaluated in *G. pectinatus* copepodites stage V. The comet assay was performed following the methods described by *Singh et al. (1988)* with modification for copepod cell extractions (*Pavlaki et al., 2016*; *Tartarotti et al., 2013*). Each treatment consisted of two replicates of 120 individuals. The positive control was a 15 min exposure to UV-C radiation (*Gong et al., 2013*; *Han et al., 2014*; *Richa, Sinha & Häder, 2015*; *Tartarotti et al., 2013*), and the negative control used filtered artificial salt water. Cell suspensions were prepared by homogenizing copepods in a Potter-Elvehjem glass homogenizer with 800 μL phosphate-buffered saline. Heavy materials from the homogenization were let to precipitate before transferring the cell suspension to 600 μL micro tubes. Cells were centrifuged at 1.0 rcf for 8 min, the supernatant was discarded, and the cell pellet was resuspended in 85 μL Low Melting Point agarose (LMP) mix (0.65% LMP in TAE 1X). Cell viability was then evaluated using hematoxylin stain to identify number of living cells. Acceptable samples contained around 30,000 cells $\cdot$ mL$^{-1}$. 40 μL of cell resuspension was added to the slides previously coated with 1% normal melting point agarose, and cooled on ice for 10 min. A second layer of 0.65% LMP was added and left to solidify before placing the slides in a lysis buffer solution overnight (2.5 M NaCl, 100 mM Na2EDTA, 10 mM Tris–HCl, 10% DMSO, 1% Triton ×100, pH 10). Slides were then rinsed with cold deionized water and submerged in an electrophoresis buffer (10 M NaOH, 200 mM Na2EDTA, pH >13) for 15 min to allow for DNA unwinding. Electrophoresis followed for 15 min at V43 and 300 mA in a horizontal gel electrophoresis tank (Bio-rad Power Pac basic cat. no. 164-5050). Slides were then rinsed three times in neutralizing buffer (0.4 M Tris–HCl), followed by dehydration in 100% ethanol for 20 min at 4 °C. Slides were stained with 50 μL SafeRed (3 μL/50 mL TAE 1X), and cell counts were done with a fluorescence microscope with a 200× magnification lens. Comet Assay VI Image analysis system, (Perspective Instruments, Wilshire, UK) (V4.3.2) was used to determine tail DNA%, tail length and tail moment in treated cells.

## Chemical analyses

Elutriate samples were digested in two percent HNO$_3$ until analyzed through Inductively Coupled Plasma Mass Spectrometry (ICP-MS Agilent 7500cx; Agilent, Santa Clara, CA, USA) for metal analysis. Polycyclic aromatic hydrocarbons (PAHs) in elutriate were analyzed by liquid extraction followed by GC-MS SIM by a commercial analytical laboratory (Hill Laboratories, Hamilton, NZ). All sediment samples were dried for 24 h at 35 °C and sieved (<2 mm fraction). Metal samples were digested in nitric acid and analyzed with ICP-MS, following US EPA 200.2 protocol. PAH concentrations were analyzed by sonication extraction, solid phase extraction clean up and GC-MS SIM analysis (US EPA 8270C). All organic compound results were normalized to 1% organic carbon to allow for direct comparison with the sediment and water quality guidelines (*ANZECC, 2000*).

**Table 1 Mean concentrations (mg · kg⁻¹ dry weight) of trace metals in sediment samples.**

| | Arsenic | Cadmium | Copper | Lead | Zinc |
|---|---|---|---|---|---|
| ANZECC ISQG-Low | 20 | 1.5 | 65 | 50 | 200 |
| ANZECC ISQG-High | 70 | 10 | 270 | 220 | 410 |
| Humber Drain | 4.7 | 0.32 | 16.7 | 32 | 310* |
| Humber Estuary | 3.7 | 0.018 | 4.7 | 9.9 | 57 |
| Old Tutaekuri Riverbed | 4.7 | 0.127 | 13.9 | 16.1 | 280* |
| Old Tutaekuri Estuary | 3 | 0.024 | 4.4 | 7.3 | 47 |
| Waitangi Estuary | 6.4 | 0.079 | 11.9 | 15.3 | 61 |

**Notes:**
Analyses from the four sites of the Ahuriri Estuary, and from the reference site, Waitangi Estuary. The ANZEEC Interim Sediment Quality Guidelines (ISQG) represent levels above which there is a low probability of biological effects (ISQG-Low) and high probability of biological effect (ISQG High). Analyses above thresholds values are indicated as * for ISQG-low.

## Statistical analysis

Results were normalized to percent of control. Homogeneity of variance was tested using Levene's test ($p < 0.05$). One-way ANOVA with Dunnet's post hoc test was conducted to identify differences between treatments and controls (*ISO, 2015*; *Stringer et al., 2014*). Comet assay raw data was log transformed for normality. Data meeting normality assumptions was analyzed with one-way ANOVA and Dunnet post hoc test. For data with variance not homogenized after log transformation (Levene's test $p < 0.05$), non-parametric Kruskal–Wallis test with a Dunn's multiple comparison test were used (*Pellegri, Gorbi & Buschini, 2014*). The R software drc, dplyr and car packages were used for the analyses and graphs.

# RESULTS

## Sediment toxicity

Only the zinc concentrations of the Old Tutaekuri riverbed and at the Humber Street drain sediment samples were above the ANZECC interim sediment quality guideline (ISQG) low value (Table 1). Acenaphthylene and acenaphthene were the only PAHs above the ANZECC ISQG-low values at the Humber Estuary site (Table S1). Elevated levels of total recoverable phosphorus were found in all samples with a much higher level at the Humber drain site (Table 2). Ammonia levels were within normal limits for all sites (*ANZECC, 2000*; *Batley & Simpson, 2009*), suggesting no masking influence on the toxicity of the samples.

All bioassays parameters met the acceptability criteria (adult survival >80%, pH 8 ± 0.2, salinity 30 ± 1 ppt, DO ≥ 8, temperature 20 ± 2 °C) (ISO 14669). There was no difference between the control groups except for potential offspring with the Waitangi Estuary sample in the *Quinquelaophonte* sp. tests ($p < 0.001$; Fig. 2). There were significant differences between the Humber Drain, Humber Estuary and Old Tutaekuri riverbed sites and the control groups for the survival and reproduction success endpoints ($p < 0.001$; Fig. 2).

**Table 2 Sediment physicochemical properties from the five study sites.**

| | pH | Redox (mV) | Ammonia (mg · kg⁻¹ dry weight) | TOC | TRP (mg · kg⁻¹ dry weight) | TN (g · 100 g⁻¹ dry weight) | Sal (ppt) |
|---|---|---|---|---|---|---|---|
| Humber Drain | 7.87 | 145 | 2.9 | 2.50 | 1,160 | 0.12 | 29.6 |
| Humber Estuary | 7.84 | 228.5 | 4.4 | 0.36 | 360 | 0.05 | 29.6 |
| Old Tutaekuri Riverbed | 7.77 | 109.0 | 22.0 | 0.15 | 340 | <0.05 | 29.6 |
| Old Tutaekuri Estuary | 7.87 | 171.4 | 9.0 | 1.06 | 390 | 0.11 | 29.6 |
| Waitangi Estuary | 7.99 | 126.0 | 5.7 | 1.45 | 650 | 0.21 | 29.6 |

**Notes:**
Values for pH and redox are the average between results from beginning and end of test.
TOC, Total Organic Carbon; TRP, Total Recoverable Phosphorus; TN, Total Nitrogen; Sal, Salinity.

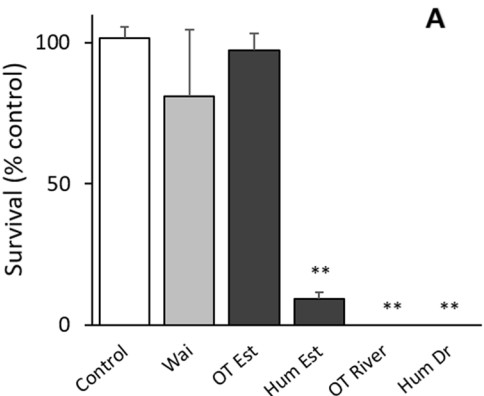

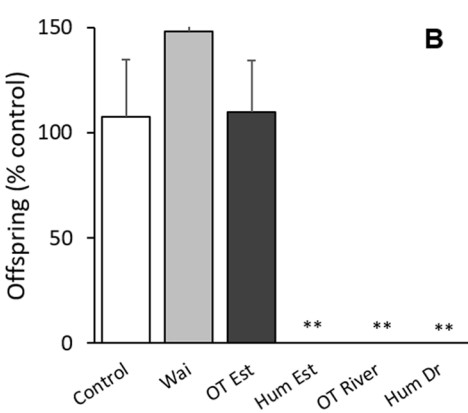

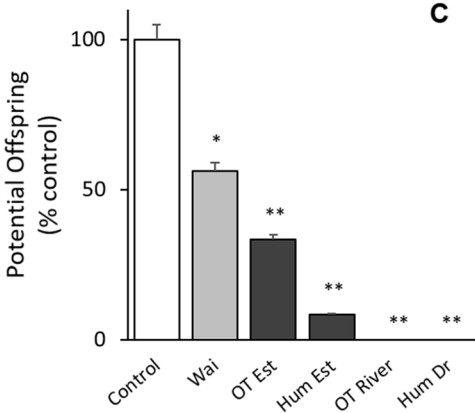

**Figure 2 Results of the 14-day chronic test using the benthic copepod *Quinquelaophonte* sp. exposed to sediments.** Three endpoints were assessed for the toxicity of sediments from the Ahuriri Estuary and reference site, Waitangi estuary. (A) Survival. (B) Total offspring hatched during the test. (C) Potential offspring (eggs in female pouch). Site codes: Waitangi Estuary (Wa), Old Tutaekuri Estuary (OT Est), Humber Estuary (Hum Est), Old Tutaekuri Riverbed (OT River), Humber Drain (Hum Dr). Diagonal lined bars: control site (Waitangi), white bar: laboratory control sediment, black bars: tested sites. Significance levels: '*' $p < 0.05$, '**' $p < 0.001$ Dunnett's multiple comparison against the control.

**Table 3 Dissolved concentrations ($\mu g \cdot L^{-1}$) of trace metals from sediment elutriates.**

| | Arsenic ($\mu g \cdot L^{-1}$) | Cadmium ($\mu g \cdot L^{-1}$) | Copper ($\mu g \cdot L^{-1}$) | Lead ($\mu g \cdot L^{-1}$) | Zinc ($\mu g \cdot L^{-1}$) |
|---|---|---|---|---|---|
| ANZECC Guidelines 99%[a] | ID | 0.7 | 0.3 | 2.2 | 7 |
| ANZECC Guidelines 95%[a] | ID | 5.5 | 1.3 | 4.4 | 15 |
| Humber Drain | <4 | <0.2 | <1 | 4.1* | 75* |
| Humber Estuary | ND | 0.2 | 1 | 1.6 | 33* |
| Old Tutaekuri Riverbed | <4 | <0.2 | <1 | 5.3* | 59* |
| Old Tutaekuri Estuary | ND | 0.2 | 2.2* | 2.2 | 44* |
| Waitangi Estuary | <4 | 0.5 | 8.3* | 4 | 33* |

**Notes:**
Analyses obtained from the four study sites at the Ahuriri Estuary, and from the reference site, the Waitangi Estuary. Analyses above the ANZECC guidelines thresholds values are indicated with * for a level of protection of 95% and above.
ID, Insufficient data; ND, Not detected.
[a] *ANZECC (2000)* water quality guidelines trigger values for 99 and 95% of protection levels for species in marine waters.

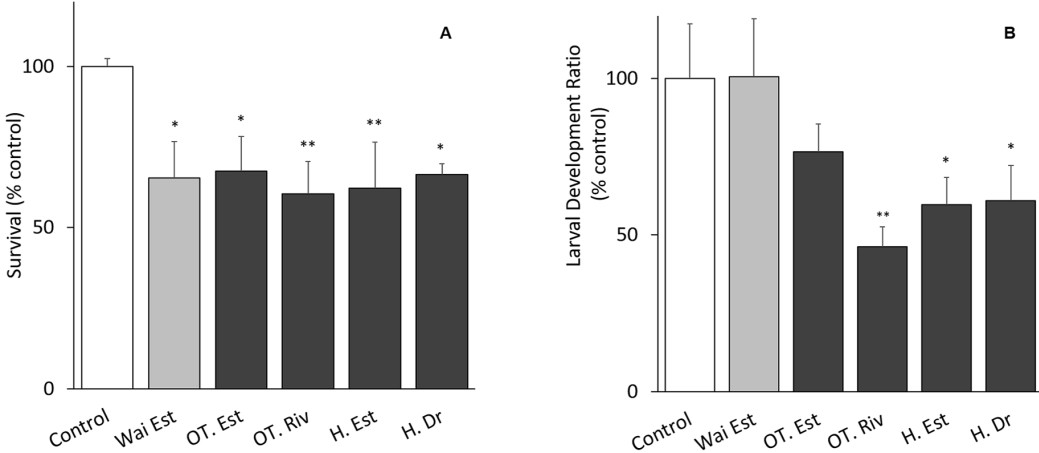

**Figure 3 Elutriates acute and chronic results.** (A) Survival and (B) larval development ratio of *Gladioferens pectinatus* after exposure to elutriates from selected sites at the Ahuriri Estuary, and Waitangi Estuary. Significance levels: '*'$p < 0.05$, '**'$p < 0.001$, Dunnett's multiple comparison against the control.

## Water column toxicity

Many of the sediment elutriate samples had dissolved trace metals at levels above ANZECC water quality guidelines except for cadmium (Table 3). All sites, including the reference site (Waitangi Estuary), had zinc levels above ANZECC trigger values for a level of protection of 95%. Copper level at the Waitangi site was also above the ANZECC water quality guideline.

The tests met the ISO 16778 (2016) quality assurance standards. Nauplii survival in samples from all sites was significantly different from the control, especially at the Old Turaekuri Riverbed site and the Humber Estuary (Fig. 3A). There were no differences in the larval development between the control, the Waitangi Estuary site and the Old Turaekuri Estuary elutriates ($p > 0.05$; Fig. 3B). The Old Tutaekuri Riverbed elutriate was the most toxic to *G. pectinatus* larval development ($p < 0.01$; Fig. 3B).

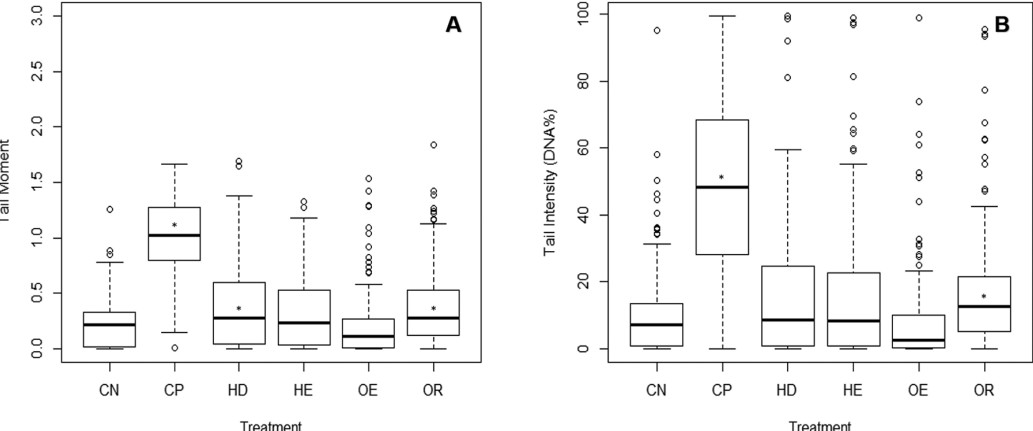

**Figure 4 DNA damage on cells isolated from copepods exposed to sediment elutriates.** (A) Tail moment. (B) Tail intensity. (CN) Negative control, (CP) positive control, and elutriates from the (HD) Humber Drain, (HE) Humber Estuary, (OE) Old Tutaekuri Estuary and (OR) Old Tutaekuri Riverbed sediments. * Indicates significance difference from control ($p < 0.05$). Humber drain ($p < 0.01$) and Old Tutaekuri Riverbed ($p < 0.001$).

## Comet assay

The comet assay was not done on the Waitangi Estuary as it was not sampled at the same time as the other sites. Test acceptability was based on the statistically significant difference between a negative and a positive control ($p < 0.001$). A significant difference was identified between the DNA damage from the negative and the positive controls (Dunn's, $p < 0.05$) on the three parameters measured (tail moment, intensity and length). There were no differences in tail moment and tail intensity between the negative control and the estuary sites. The Old Tutaekuri Riverbed samples induced DNA damage as indicated by the tail moment and tail intensity (DNA%) (Fig. 4). There was no significant DNA damage in samples from Humber Drain based on tail intensity endpoint, however results from tail moment suggest a marginal significance ($p = 0.06$).

## DISCUSSION

Between 79% and 87% of runoff loads have been attributed to urban runoff (residential galvanized roofs and urban roads), with 86% of the load containing zinc, especially after wet events (*Davis & Birch, 2010*; *Li et al., 2015*). Metal concentrations observed in this study were comparable to earlier reports at the Ahuriri Estuary (*Smith, 2014*), and studies in urbanized New Zealand cities (*Charters, Cochrane & O'Sullivan, 2016*; *Mills, 2016*) and overseas (*Ács et al., 2016*). The results of metals in the systems studied suggest that stormwater runoff is a major source of metals in the estuary, specially at the Old Tutaekury Riverbed, and Humber Drain samples.

The high mortality and low reproduction of *Quinquelaophonte* sp. in the Humber Estuary sample could not be fully explained by the individual metal or PAH content (except for acenaphthylene and acenaphthene which were above the ISQG-low values). This suggests the presence of multiple stressors affecting the toxicity in copepods. In this study, the Humber Estuary site had the highest sum of low molecular weight PAHs

(Table S1). Recent studies have shown that copepods can significantly bioaccumulate PAHs, even when these are below threshold values (*Cailleaud et al., 2009*). However, low weight PAHs can be eliminated more efficiently than high molecular PAHs due to their lower hydrophobicity nature (*Almeda et al., 2013*). This could have partially contributed to the high mortality of copepods (91%) exposed to the Humber Estuary sample, and the unsuccess in egg hatching. Our study aligns with those conducted by *Boehler et al. (2017)* and *Heinrich et al. (2017)* on extracts from sediment samples collected near the Ahuriri Estuary. They showed multiple toxicity responses including androgenic and glucocorticoid activities in cell lines, and moderate teratogenicity in zebrafish embryos exposed to PAHs and musk extracts mixtures at concentrations as low as 1.6 mg SEQ $\cdot$ mL$^{-1}$.

The total mortality of copepods in the tributary sites can be partially explained by the concentration of zinc. *Stringer et al. (2014)* calculated a zinc LC$_{50}$ of 196 mg kg$^{-1}$ and a mobility inhibition EC$_{50}$ of 137 mg kg$^{-1}$ for *Quinquelaophonte* sp. The results from the drain and riverbed samples were well above that value (310 and 280 mg kg$^{-1}$ respectively), suggesting that zinc had an effect on the mobility and mortality of *Quinquelaophonte* sp. individuals during the 14-day exposure.

Physicochemical results showed that the Humber Estuary samples had higher redox potential, which can trigger sulphides and trace metal sulphides oxidation increasing metal bioavailability in pore water (*Chapman et al., 1998*; *Kalnejais, Martin & Bothner, 2015*; *Lu et al., 2016*). In addition, the presence of mucilage (predominantly from diatom) was also observed to cover the sediment samples from the Humber Estuary site at the end of the tests. Mucilage has been reported to affect copepod's grazing ability (*Malej & Harris, 1993*; *Pančić & Kiørboe, 2018*), and cause teratogenic effects in the gonadal tissue, affecting reproductive success (*Ianora et al., 2011*; *Wolfram, Nejstgaard & Pohnert, 2014*). Further studies are needed to confirm the contribution of these stressors to the toxicity.

Bioassays using *G. pectinatus* have been confirmed as good indicators for the toxicity of water dissolved compounds through contact or ingestion. Despite of the reduction of metal concentrations from the tributaries to the estuary sites (zinc and lead dropping by 2.27× and 2.5× respectively), survival and larval development were significantly affected. The toxicity of the sediment samples to *G. pectinatus* is unlikely to be solely due to independent metals, as the levels from the estuary sites were similar to those measured in the Waitangi reference and yet, there were differences between the larval development ratio. The presence of other compounds not addressed in this paper (i.e., organochlorine compounds and emerging contaminants) may have further influenced the toxicity in *G. pectinatus,* as reported in *Boehler et al. (2017).* This may be due to the higher exposure to pollutants emerging from the Napier city, while the Waitangi estuary is distanced from the urban stormwater and waste water runoff. Extensive literature has reported the effect of additive toxicity of chemicals mixtures in marine invertebrates reproductive success and mortality (*Cooper, Bidwell & Kumar, 2009*; *Hagopian-Schlekat, Chandler & Shaw, 2001*; *Picone et al., 2018*). However, these effects highly depend on the type of interaction (metal-metal, metal-organic, organic-organic) and on the concentration of each compounds in the mixture, which can either trigger synergistic, antagonistic, or
non-interactive toxicities (*Gauthier et al., 2014*; *Nys et al., 2016*; *Traudt, Ranville & Meyer, 2017*). A recently published framework for ecological risk assessment (*Nys et al., 2018*) was developed to better characterize the bioavailability of metal mixtures, based on a sensitive species distribution model. This approach could be further addressed when assessing community impacts at the Ahuriri estuary.

The comet assay is a well recognized method to measure DNA damage in individual cells, and an accepted tool for environmental monitoring (*Colin et al., 2016*; *de Lapuente et al., 2015*; *Martins & Costa, 2015*). The present study confirms the genotoxicity in the Old Tutaekuri samples, and the potential genotoxicity at the Humber drain, previously characterized with sediment extracts of organic compounds from the Ahuriri estuary (*Boehler et al., 2017*; *Heinrich et al., 2017*). The Humber Drain and Old Tutaekuri Riverbed sites had zinc and lead concentrations above guidelines values. Zinc has been reported to induce low levels of DNA damage in zooplankton copepods (*Goswami et al., 2014*), the marine clam *Ruditapes philippinarum,* rainbow trout *Oncorhynchus mykiss*, and grasshopper *Chorthippus brunneus* (*Augustyniak et al., 2006*; *Marisa et al., 2016*). However, while lead genotoxicity has not been sufficiently studied in copepods, it has been reported to be significant in amphipod haemocytes and spermatozoa at 25 $\mu$g $\cdot$ L$^{-1}$ (*Di Donato et al., 2016*), in freshwater mussel at 120 $\mu$g $\cdot$ L$^{-1}$ (*Black et al., 1996*; *Sohail et al., 2017*) and in polychaetes at 100 $\mu$g $\cdot$ L$^{-1}$ after three days (*Singh, Bhagat & Ingole, 2017*).

In addition, multiple co-genotoxic mechanisms between chemical compounds have been reported on aquatic organisms (*Gauthier et al., 2014*). These mechanisms include the inhibitory effect of metals on the cytochrome P450, essential for PAHs detoxification; the enhanced production of reactive oxygenated species which translates into cellular oxidative stress, and the effects of PAH on membrane integrity, which enhances the permeability to metals. This indicates that mixtures of organic and inorganic chemicals can modulate synergistic, additive or antagonistic toxicities at the cellular and genetic level (*Di Poi et al., 2018*; *Kousar & Javed, 2015*; *Warne & Hawker, 1995*).

The initial protective mechanism of copepods to minimize cellular and DNA damage is to metabolize and eliminate toxicants (*Han et al., 2017*). However, in the presence of toxicants with pharmacological modes of action (non-specific disruption or overloading of enzyme activities), the enzymes in charge of metabolizing compounds and lipid storage regulators may be inhibited. The presence of compounds with these modes of action were identified at the Humber drain and the Old Tutaekuri riverbed in previous studies at 0.012 and 0.0034 mg kg$^{-1}$ respectively (*Boehler et al., 2017*). Pharmacological effects have been reported in the pelagic copepod *Calanus finmarchicus* exposed to naphthalene, pyrene, crude oil (*Han et al., 2015*; *Hansen et al., 2008*), *A. tonsa* exposed to four synthetic musk substances (*Wollenberger et al., 2003*), *Tigriopus japonicus* exposed to polybrominated diphenyl ethers (*Han et al., 2015*), and *Nitocra spinipes* exposed to galaxolide (*Breitholtz, Wollenberger & Dinan, 2003*). This may suggest that the chemicals present at the impacted estuary sites may have modes of action affecting the precursor of larval molding and deoxidation (P450 enzymes such as CYP330A1 or CYP305A1) as shown on the bioassays, as opposed to directly fractioning DNA brands (*Hansen et al., 2008*). Further, DNA repairing mechanisms present in copepods remains to be confirmed.

It has been suggested that risk assessment frameworks and decision support systems consider the genotoxicity of chemical mixtures as part of a weight of evidence approach. This has been demonstrated by the method integrating nine lines of evidence in contaminated estuaries in Portugal (*Caeiro et al., 2017*), the policy frameworks and risk assessment tools at contaminated sites in the Netherlands (*Dagnino et al., 2008*), and the acceptance of genotoxic tools as one of the standard monitoring strategies within the European Union Water and Marine Strategy Framework Directives (*Allan et al., 2006*; *Martins & Costa, 2015*).

## CONCLUSION

The sediment and elutriate samples from the urban Ahuriri Estuary tributaries were genotoxic and impacted reproduction in the two copepod bioassays. The toxicity from the estuaries sites was lower than that of the tributaries, but genotoxicity and reproduction inhibition were evidenced. This confirms the bioavailability of biologically active chemicals in the sediment samples. The approach using bioassays in pelagic and benthic copepod species was successful to characterise the toxicity of urban sediment. The results provides useful effects-based information to inform environmental managers.

## ACKNOWLEDGEMENTS

The authors would like to thank Anna Madarasz-Smith for the collection of sediment samples and for the access to chemical data; Olivier Champeau for assistance with the comet assay; and Steve Web for his help with copepod histology and cells identification.

### Funding
This work was supported by the Hawke's Bay Regional Council, Callaghan Innovation R&D Student Fellowship Grant (No. BMISK1401), and Boffa Miskell. The co-supervisor of the project is a senior ecologist from Boffa Miskell. He assisted in the study design.

### Grant Disclosures
The following grant information was disclosed by the authors:
Hawke's Bay Regional Council, Callaghan Innovation R&D Student Fellowship: BMISK1401.

### Competing Interests
Vaughan Keesing is a senior ecologist employee at Boffa Miskell, one of the funding contributors to the study. Mark Costello is an Academic Editor for PeerJ.

### Author Contributions
- Maria P. Charry conceived and designed the experiments, performed the experiments, analyzed the data, contributed reagents/materials/analysis tools, prepared figures and/or tables, authored or reviewed drafts of the paper, approved the final draft.
- Vaughan Keesing authored or reviewed drafts of the paper.

- Mark Costello conceived and designed the experiments, authored or reviewed drafts of the paper, approved the final draft.
- Louis A. Tremblay conceived and designed the experiments, contributed reagents/materials/analysis tools, authored or reviewed drafts of the paper, approved the final draft.

## Data Availability

The raw data are provided in the Supplemental Files.

## Supplemental Information

Supplemental information for this article can be found online at http://dx.doi.org/10.7717/peerj.4936#supplemental-information.

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
