# Peer review of "Assessment of the ecotoxicity of urban estuarine sediment using benthic and pelagic copepod bioassays"

_PeerJ, doi:10.7717/peerj.4936_

## Round 0.1 · original submission · Major Revisions

The paper represents a valuable contribution to the area of aquatic ecotoxicology. Besides a number of corrections suggested by the reviewers, it is necessary to update the literature survey including recent publications and improving discussion.

Reviewer 1 ·

Basic reporting

This study brings a contribution for aquatic ecotoxicology using copepod bioassays in estuarine sediment biomonitoring.

Experimental design

No comment.

Validity of the findings

No comment.

Additional comments

The term "in situ" is not applicable to this study, because the tests were carried out in laboratory. Please verify the use of this term in the text (e.g., L 59-60) and in the abstract (L. 30).
Methodology:
- How long the samples were stored until the tests were carried out? (L. 88).
- Please provide details about elutriate preparation (L. 94).
- Please provide the author of the species cited in this study.
- What is the source of "mixed algae"? Which species? (L.112)
- Please cite the protocol that was followed to cultivate the aquatic organisms in lab (L. 103-113).
- Please provide more details about the sediment tests to allow the reproductibility of this study by other researches (L 114-121).
- Please standardize the terms. E.g., "waterbourne toxicity tests" (L. 123) should be replaced by "elutriates toxicity tests".
- These references are not provided in the reference section: HBCR 2014 (L. 50), ASTM 2000 (L. 125), USEPA & USACE 1998 (L. 125), ISO 16778 (L. 133), ISO 2016 (L. 199).
- What the term "development rations" means? (L. 142).
- Please provide a reference to the statistical analysis (L. 171-178).
- Tables 1 and 2 are exchanged in the text (L. 182 and 186).
- L. 115, please complete “Chandler & Green” (1996).
- L. 196, "arsenic" or "cadmium"?
- L. 252-254. Please clarify. Why lower concentrations of metals induced higher DNA breaks than higher concentrations?
- What is the reason for effects on survival but not in development?

·

Basic reporting

The study presented in the manuscript can be considered in correct English. However, line 21, the species name is misspelled; line 38, should be lives; line 123 is waterborne.

The Literature references are adequate (n = 70), however there are many studies with more than 5 years (<2013, n = 47) and few current studies (≥ 2013, n = 23).

I believe that “water column toxicity” is not the correct term, due to the fact that you are working with elutriate (line 194).

Experimental design

In the material and methods section, line 111 to 113 (test species), you should improve the description of the protocol for cultivation of these species in the laboratory (e.g.: photoperiod; room's conditions or hatchery equipment used; mixed algae: which species,...). Some of them, you describe later in the toxicity test section, but are missing in the test species section.

Validity of the findings

Please check, I believe the order of tables 1 and 2 have been changed, as well as table S1.

Lines 199 to 202, you started describing the results of Fig 3B and after Fig. 3A. I believe you should change the order of the graphs in the figure or in the text, to be clearer for the reader.

Line 202, please add the letter "B" when you mentioned "Fig. 3" (Fig. 3B).

Line 210, you mentioned that a significant difference was identified at some endpoints of DNA damage. I think the tail length graph is missing.

Also in line 210, you mentioned that "there were no difference between the tail moment from the negative control and the estuary sites" (Fig. 4). I believe you should also include the tail intensity.

However in line 212, please check, but I believe you could not affirm that the Humber Drain samples induced DNA damage as indicated by the tail intensity.

In table 3, "*" is missing in the concentration of Zinc of Humber Drain (75 μg/L).

In the discussion section, you mentioned high and low concentration and effects of other studies to discuss your results. However, you should include the results of those studies to determine or to be clearer what you consider as high or low values.

Additional comments

The study presented in the manuscript can be considered relevant. Although, the references used in the introduction and discussion may be more current.

The study also indicated that further studies are needed to conclude which chemical compounds were most influential in the toxic effects on the copepods of the estuarine sediment. Since the chemical compounds analyzed did not fully answer this question. Due to the fact that the reference area presented some metals and the toxicity was detected in some bioassays, this should have been better discussed in the text.

---

## Round 0.2 · accepted · Accept

The improvements and corrections required by the reviewers as well as the recommendations made by the Editor were included in the revised version. Therefore, the decision is to accept the manuscript for publication in its current format.

#